# Rationale for 1068 nm Photobiomodulation Therapy (PBMT) as a Novel, Non-Invasive Treatment for COVID-19 and Other Coronaviruses: Roles of NO and Hsp70

**DOI:** 10.3390/ijms23095221

**Published:** 2022-05-07

**Authors:** Lydia C. Kitchen, Marvin Berman, James Halper, Paul Chazot

**Affiliations:** 1Department of Biosciences, Durham University, Durham DH1 3LE, UK; lydia.kitchen@durham.ac.uk; 2Quietmind Foundation, Philadelphia, PA 19147, USA; marvinberman@quietmindfdn.org (M.B.); james.halpermd@gmail.com (J.H.)

**Keywords:** SARS-CoV-2, COVID-19, photobiomodulation, viral replication, inflammation, 1068 nm, nitric oxide, thrombosis, cytoprotection, Hsp70

## Abstract

Researchers from across the world are seeking to develop effective treatments for the ongoing coronavirus disease 2019 (COVID-19) outbreak, which arose as a major public health issue in 2019, and was declared a pandemic in early 2020. The pro-inflammatory cytokine storm, acute respiratory distress syndrome (ARDS), multiple-organ failure, neurological problems, and thrombosis have all been linked to severe acute respiratory syndrome coronavirus 2 (SARS-CoV-2) fatalities. The purpose of this review is to explore the rationale for using photobiomodulation therapy (PBMT) of the particular wavelength 1068 nm as a therapy for COVID-19, investigating the cellular and molecular mechanisms involved. Our findings illustrate the efficacy of PBMT 1068 nm for cytoprotection, nitric oxide (NO) release, inflammation changes, improved blood flow, and the regulation of heat shock proteins (Hsp70). We propose, therefore, that PBMT 1068 is a potentially effective and innovative approach for avoiding severe and critical illness in COVID-19 patients, although further clinical evidence is required.

## 1. Introduction

In December 2019, reports of the novel infectious coronavirus disease 2019 (COVID-19) caused by the severe acute respiratory syndrome coronavirus 2 (SARS-CoV-2) emerged in Wuhan, China. By the end of February 2020, SARS-CoV-2 had spread to 51 countries and, with the rapid growth in case numbers crippling all aspects of life globally, the situation was declared a pandemic by the World Health Organisation (WHO) on 11 March 2020. As of 27 April 2022, there have been over 508 million confirmed cases of COVID-19 worldwide, with 6.2 million deaths (https://covid19.who.int (accessed on 27 April 2022)), giving rise to an urgent need for effective and safe treatment strategies.

Due to the pro-inflammatory and pro-thrombotic nature of COVID-19, photobiomodulation therapy (PBMT) could be used to modulate the immune and thrombotic system and repair damaged host tissues. Here, we review the specific use of 900–1068 nm near infrared (NIR) light in the treatment of patients infected with SARS-CoV-2, including direct and indirect antiviral effects.

## 2. Structure and Replication of SARS-CoV-2

The newly identified SARS-CoV-2 is an enveloped, non-segmented, positive sense ssRNA virus with a diameter of 65–125 nm (Figure 1). The virus is comprised of four strutural proteins: spike (S), small envelope (E), membrane (M), and nucleocapsid (N), alongside several accessory proteins [1,2]. SARS-CoV-2 is a β-coronavirus of the Coronavirinae family that also contains the SARS-CoV and MERS-CoV viruses, responsible for the 2003 Severe Acute Respiratory Syndrome (SARS) and 2012 Middle East Respiratory Syndrome (MERS) outbreaks, respectively [1]. At an early stage of the current outbreak, genome analysis identified 79.6% sequence similarity between SARS-CoV-2 (then named 2019-nCoV) and SARS-CoV, hence the name SARS-CoV-2 [3].

SARS-CoV-2 is a highly infectious respiratory pathogen, with human transmission occurring through airborne respiratory droplets from coughs and sneezes, and surface contamination [4,5,6]. There is evidence that SARS-CoV-2, like SARS-CoV, uses the host angiotensin-converting enzyme 2 (ACE2) receptor as an entry point into cells; SARS-CoV-2 is able to enter HeLa cells expressing ACE2 from a number of organisms, including humans, but is unable to enter the untransfected cells [3]. The normal function of ACE2 is in the control of blood pressure, catalysing the hydrolysis of angiotensin II into the vasodilator angiotensin (1–7) [7]. Thus, ACE2 is expressed in a number of cell types around the body. The immunohistochemistry of human tissue samples has shown that ACE2 receptors are found in lung alveolar epithelial cells, nasopharyngeal and oral mucosa, the endothelium, the brain, the gastrointestinal tract, and in peripheral organs, e.g., the liver and kidneys [8,9,10].

Figure 2 shows the mechanism of SARS-CoV-2 entry into a target cell. Upon the binding of the viral S protein to the host ACE2, the host cell TMPRSS2 cleaves the S peptide, activating the S2 domain and driving membrane fusion between the viral envelope and host cell membrane. Both ACE2 and TMPRSS2 are highly expressed in alveolar epithelial type II cells [11], explaining why COVID-19 typically affects the lungs the most. The virus is then able to release the genomic material into the host cell cytoplasm, where the ssRNA acts as mRNA to be translated and produce the viral replicative enzymes. The N proteins assist in translation by binding tightly to the RNA, making it more accessible to host ribosomes [2]. The new viral proteins assemble, forming small vesicles to be exported out of the cell by exocytosis, and allow the new virions to spread around the body.

## 3. COVID-19 Symptoms and Complications

COVID-19 can manifest in a variety of ways: many individuals infected with SARS-CoV-2 remain asymptomatic, but severe complications are seen in other patients [12]. The most common symptoms are fever, cough, and fatigue, mirroring those of SARS and MERS. In some patients, muscle pain, sputum production, headache, haemoptysis, dyspnoea, and diarrhoea, among others, may present as symptoms [6,13]. With increasing disease severity, complications such as acute respiratory distress syndrome (ARDS), hyperinflammation, organ failure, and death become more prevalent [6,13,14]. A Chinese study of over 44,000 confirmed COVID-19 cases found 81% were defined as mild to moderate (from asymptomatic to mild pneumonia), 14% were severe (dyspnoea, blood oxygen saturation <93%, and/or lung infiltrates >50%), and 5% were critical (respiratory failure and/or multiple organ dysfunction or failure) [15], although these percentages vary with time and location.

Though COVID-19 was initially thought to be just a viral pneumonia, SARS-CoV-2 infection can, in fact, lead to multiple organ dysfunction [6,8]. The ability of SARS-CoV-2 to target multiple organs has been attributed to a combination of widespread ACE2 distribution and systemic cytokine storms [16,17]. A cytokine storm is an uncontrolled inflammatory response due to an excessive release of pro-inflammatory cytokines by the host’s immune system [17,18]. The initial release of cytokines is a form of defence against SARS-CoV-2 by the innate immune system, but when production becomes excessive in critically ill patients, it may cause a serious pro-inflammatory condition [19]. Many cytokines show elevated levels in COVID-19 patients, including interleukins (IL-1β, IL-2, IL-7, IL-8, IL-9, IL-10, IL-17) and tumour necrosis factor alpha (TNF-α) [6,20]. The systemic inflammation caused by increases in serum and plasma cytokine levels has been linked to both disease severity and the likelihood of ARDS [17,21], and the cytokine storm remains the key cause of COVID-19 deaths. Therefore, harnessing the immune system to reduce this exaggerated inflammation could be vital to effectively manage patients with COVID-19 [22].

Thrombosis is emerging as a significant contributor to COVID-19 mortality. COVID-19 patients, especially those in the intensive care unit (ICU), show a high incidence of hypercoagulability in the form of venous and arterial thromboembolism. The most common coagulation events in COVID-19 patients are pulmonary embolisms (PEs) [23,24], which are blood clots in the lungs. PEs can harm the lungs by restricting blood flow, lowering blood oxygen levels, and affecting other organs. Large or multiple blood clots can be fatal. Other thrombotic complications of COVID-19 include venous thromboembolism (VTE), deep-vein thrombosis, ischemic stroke, myocardial infarction, and microvascular thrombosis [23,25].

### 3.1. Neurological Symptoms

A wide variety of neurological manifestations are being increasingly observed in COVID-19 patients [26,27,28]. A retrospective case study of 214 hospitalised patients with confirmed SARS-CoV-2 infection discovered that 36.4% displayed neurological symptoms [28], and a 6-month study of the medical records of 236,379 COVID-19 survivors showed an incidence rate for neurological and psychiatric diagnoses of 33.6% [27]. Anosmia and ageusia (loss of smell and taste, respectively) are particularly prevalent, with a Korean study observing these symptoms in 15.3% of 3191 patients with early-stage COVID-19. Of the patients exhibiting these symptoms, the majority (79.6%) had asymptomatic to mild disease severity [29]. COVID-19 symptoms of the central nervous system (CNS) include impaired consciousness, headache, acute cerebrovascular disease, ischemic stroke, encephalopathy, delirium, and seizures. Effects on the peripheral nervous system (PNS) include loss of smell and taste, nerve pain, and Guillain–Barré syndrome [28,30,31]. Mao et al. (2020) found that increased severity of COVID-19 increases the likelihood of CNS symptoms, with significant increases in acute cerebrovascular disease and impaired consciousness. Evidence is emerging of psychiatric complications of COVID-19, particularly mood and anxiety disorders [26,32]. A study of 103 COVID-19 patients compared with 103 matched controls found that those with COVID-19 had higher levels of depression, anxiety and post-traumatic stress symptoms (*p* < 0.001) [33].

There are two pathways by which SARS-CoV-2 may infect the CNS: the hematogenous route or the neuronal route [30,34,35,36]. Through the hematogenous route, the virus that has infected the lower respiratory tract infects the endothelial cells of the lung capillaries, followed by astrocytes and macrophages. The virus enters the blood stream and is transported to the blood–brain barrier (BBB). The respiratory virus can damage the BBB endothelial cells, gaining entry to the brain [30,35,36,37]. The alternative neuronal route begins with viral presence in the upper respiratory tract [30]. The virus targets peripheral nerve endings: primarily olfactory neurons of the nasal epithelia [36,38,39], but potentially also the vagus nerve through the lung–gut–brain axis [40,41,42]. Viral pathogens can cross neurons and synapses, exploiting the motor proteins dynein and kinesin for retrograde and anterograde movement along axons [35]. The transneuronal pathway from the olfactory epithelium to the olfactory bulb and olfactory nucleus is supported by the detection of SARS-CoV-2 from nasal swabs, anosmia caused by COVID-19, and high ACE2 levels in the nose [36,43]. As SARS-CoV-2 is a novel coronavirus, the route it uses to enter the CNS has yet to be demonstrated experimentally. However, given the 79.6% sequence similarity with SARS-CoV [3], it is reasonable to assume the two viruses share similar neurotrophic mechanisms [34]. Therefore, it is possible that SARS-CoV-2, like SARS-CoV, uses both the hematogenous and neuronal routes to hijack the nervous system [37,44]. 

Baig et al. (2020) relate the neurological symptoms of COVID-19 to the expression of ACE2 in the CNS [37]. Immunohistochemistry shows that ACE2 is expressed in the endothelia and the smooth muscle cells of the brain [9]. More recent studies have found ACE2 expression in neurons and glia [45,46]. Within the neurons, ACE2 protein expression is highest in the cell body, with lower expression in the axons and dendrites. This was demonstrated using immunocytochemistry studies of human pluripotent stem-cell-derived neurons [47]. Though it is established that SARS-CoV-2 is able to infect the CNS, it remains unknown if the neurological issues observed in COVID-19 patients are due to direct viral binding to ACE2 in the brain or to the cytokine storm causing systemic hyperinflammation, including neuroinflammation [36,48].

### 3.2. Long COVID

COVID-19 is now known to cause post-infection sequelae, termed long COVID or post-COVID-19 syndrome. A WHO-led Delphi consensus defined this condition by symptoms lasting more than two months (with no other cause) in patients with confirmed or probable past SARS-CoV-2 infection [49]. Symptoms are wide-ranging in both expression and severity, affecting almost every organ system in the body, including the respiratory, nervous and cardiovascular systems. A preprint meta-analysis of 40 studies concluded that there were 100 million cases of long COVID as of November 2021, with fatigue, shortness of breath, insomnia, joint pain and memory problems being the five most common symptoms. The prevalence of long COVID is 43%, which increases to 57% if a patient required hospitalisation for acute COVID-19 [50]. As cases of COVID-19 have more than doubled to 508 million since the meta-analysis was written, it can be estimated that cases of long COVID now stand at over 200 million. With so many unknowns about the cause and mechanism of this very common syndrome, finding treatments which may work to combat the chronic and delibertating symptoms is a vital next step in the research community.

## 4. Photobiomodulation

Photobiomodulation therapy (PBMT) shows promise as a self-administered, noninvasive treatment option for COVID-19. Shortly after the 1960 discovery of the monochromatic light source [51], PBMT was accidentally discovered by the Hungarian Endre Mester in 1967 [52]. Mester was using a red laser to reduce the size of cancerous tumours in mice, but the laser had a lower power than he intended. Instead of observing changes to the tumour as predicted with the high-power laser, he noticed that the wounded skin of the laser-treated mice healed faster. The laser caused hair to grow back faster in the shaved areas and the wounds healed better, so low-level light appeared to be promoting tissue repair. Mester spent the rest of his career investigating this phenomenon, carrying out further promising experiments on wounds, skin defects, burns, ulcers and bedsores [53,54].

PBMT is defined as a light-based therapy for the stimulation, enhancement and healing of cell and tissue function. This use of low-energy light to stimulate biological effects was formally named low-level laser therapy (LLLT), although this name was later changed to the more accurate ‘photobiomodulation’. This was because light-emitting diodes (LEDs) can more safely deliver the same beneficial effects as lasers, and ‘low-level’ was considered subjective [55,56,57]. The effects of PBM appear to be limited to a specified set of wavelengths of light, most commonly the red (600–700 nm) and near infrared (NIR, 750–1300 nm) regions of the electromagnetic spectrum [56,58]. More current investigations have identified distinct optical windows within the near infrared spectrum (810nm and 1064nm) with marked differences in production of oxygenated hemoglobin and cytochrome c oxidase [59]. PBM effectiveness is also dependent on the energy dosage supplied [58], following a ‘biphasic dose response’ curve. This obeys the Arndt–Schulz law, where doses higher or lower than the optimum dose cause reduced or, in the case of very high energy levels, negative therapeutic effects via ‘bio-inhibition’ [56,60,61].

### Molecular Mechanisms of PBMT

The work of Tiina Karu from Russia has revolutionised the understanding of the molecular mechanisms of PBMT. Karu demonstrated that a mixed valence form of cytochrome *c* oxidase (CCO), the terminal unit IV enzyme of the mitochondrial electron transport chain (ETC), is the primary photoacceptor for red-NIR light in mammalian cells [62,63,64]. The identification of CCO as the photoreceptor explains the wavelengths that commonly show biological effects from PBM and allows the molecular mechanisms of PBM to be proposed (Figure 3). Often 600–700 nm and 760–900 nm (red and NIR light, respectively) are used in PBMT, and these wavelengths correspond with peaks in the CCO absorption spectrum [56,58,63]. CCO is a large enzymatic complex located within the inner mitochondrial membrane. The complex contains two copper centres (Cu_A_ and Cu_B_) and two haem centres (a and a_3_). Upon NIR irradiation, nitric oxide (NO) dissociates from the O_2_-binding site (a combination of the a3 and Cu_B_ centres) of CCO [65]. NO is inhibitory as it competes with O_2_ for the binding site, so the NO dissociation increases CCO enzymatic activity [66]. CCO oxidises cytochrome *c* and utilises the released electrons to reduce molecular O_2_. Upon binding of this reduced product to mitochondrial protons (H^+^), H_2_O is generated within the mitochondrial matrix, increasing the H^+^ gradient across the inner membrane. ATP (adenosine triphosphate) synthase uses this electrochemical potential to synthesise ATP [65,67]. Many studies, both in vitro and in vivo, have demonstrated that PBM causes an increase in intracellular ATP (reviewed in [68]). The activation of the ETC through PBM also increases reactive oxygen species (ROS), Ca^2+^ ions, and cyclic Adenosine Monophosphate (cAMP). These signalling molecules induce changes in transcription factors such as the nuclear factor kappa-light-chain-enhancers of activated B cells (NF-κB, [69]), and result in long-term cellular effects, as detailed in Section 5. 

In recent years, other wavelengths have been shown to also have beneficial biological effects, for example, 1068 nm [70] and 1072 nm [71,72,73]. Although CCO will absorb less light at these wavelengths, lower scattering means the longer wavelength light is able to travel deeper within tissues and stimulate more CCO and ion channels. The absorption of 1068–1074 nm light causes vibrations of nanostructured water, leading to the opening of calcium ion channels, such as transient receptor potential (TRP) channels [74,75]. In addition, 1068 nm NIR light generates peak transmission through water molecules, so less energy is used to enter biological materials [70,76]. For these reasons, this review focuses on 1068 ± 25 nm.

## 5. Rationale for PBMT to Treat COVID-19

PBMT has been successful in the treatment of viral infections and respiratory diseases, suggesting feasibility for the treatment of COVID-19. Low-level 1072 nm infrared light was shown to significantly reduce the time taken for HSL (herpes simplex labialis) lesions to heal compared to sham treatment [71]. Whilst the antiviral mechanisms of this therapy are not yet fully understood, some feasible explanations are explored in this section.

It has also been established that PBMT reduces lung inflammation in experimental models, including LPS-induced pulmonary inflammation in mice [77] and rats [78], and in mice submitted to cigarette smoke to mimic chronic obstructive pulmonary disease (COPD) [79]. Several small-scale, peer-reviewed studies report the benefits of PBMT on respiratory disorders in human patients, including asthma [80] and COPD [81]. Shorter recovery times, less medication reliance, fewer respiratory symptoms, and improved radiological, immunological, and blood markers are all positive outcomes of PBMT seen in these patients [22]. 

The first major clinical trial of PBMT for COVID-19 patients was carried out by Vetrici et al. [22]. Despite the low sample size, the study demonstrated that adjunctive PBMT improved the clinical status of COVID-19 pneumonia above standard medical care. PBMT (808 nm and 905 nm) applied to the lungs increased peripheral oxygen saturation, relieved pulmonary symptoms, and improved chest X-ray findings. This suggests that PBMT could be used to improve COVID-19 patients’ respiratory and clinical conditions, decreasing the requirement for ventilator support and ICU stay. Similarly, a placebo-controlled trial of thirty severe COVID-19 patients found that, whilst the length of ICU stay did not change between groups, patients treated with 905/633/850 nm PBMT-sMF (PBMT combined with static magnetic field) showed reduced diaphragm atrophy and improved ventilatory parameters and lymphocyte count [82]. A case report by Sigman et al. (2020) used a combination of 808 and 905 nm PBM to treat a 57-year-old man with a severe case of COVID-19 pneumonia. The patient’s radiological findings, respiratory rates and oxygen requirements improved significantly after treatment, with no need for the predicted ventilator treatment [83]. These clinical reports all support the use of PBMT to treat COVID-19 and reduce the pressure on health services.

Up to now, most studies in the field of PBMT have focused on 600–700 nm and 780–850 nm wavelengths, but irradiation by 1060–1080 nm light has shown significant behavioural effects including cognitive enhancement [84,85] and executive functions [86], and so would be worth investigating for the neurological, as well as immune, features of COVID-19. The next sections (5.1–5.5) explore the rationale for these longer wavelengths of PBMT to treat COVID-19.

### 5.1. Cytoprotection

Research from our laboratory has provided evidence that CAD neuronal cells exposed to 1068 nm light are significantly protected against β-amyloid_(1–42)_-induced cell death [70]. This cytoprotection is also observed in human lymphocytes treated with IR-1072, but not IR-880, indicating these higher wavelengths of light could be more useful for improving cellular viability [73]. If photobiomodulation could protect immune, pulmonary, glial, and neuronal cells from SARS-CoV-2 infection by photobiomodulation, perhaps via a photo-preconditioning mechanism, this would offer a prospective, simple, and non-invasive treatment for COVID-19, including the prevalent neurological consequences.

### 5.2. iNOS and NO

It is thought that iNOS (inducible nitric oxide synthase) is vital in a host’s immune response against pathogens and is induced in the case of inflammation or infection. In an investigation of MRSA infection in mice, 1072 nm light caused long-term changes in iNOS, with mRNA expression increasing by 2.7 times compared to control mice 5 days post-treatment [72]. Similar effects have been observed in human lymphocytes, with quantitative immunoblotting showing 4.9 times higher iNOS protein expression following IR-1072 treatment, but *not* with IR-880 [73]. PBMT, therefore, increases nitric oxide (NO) both indirectly, through an increase in iNOS expression, and directly, through the photo-dissociation of NO from the CCO enzyme. 

NO may be responsible for several of PBMT’s positive effects. As a well-known inhibitor of apoptosis, as seen in vitro [87,88,89], NO may improve the viability of various cell types against stressors such as SARS-CoV-2 infection. NO interacts with reactive oxygen and nitrogen intermediates to form a range of antimicrobial molecular species [90] which are also useful in an immune response. Most importantly, NO inhibits RNA replication in several types of viruses [91,92], including SARS-CoV [93] and SARS-CoV-2 [94]. NO targets viral proteases; in SARS-CoV-2, it is believed that the S-nitrosylation of the 3CL cysteine protease inhibits the protease cleavage of viral polyproteins [94]. NO is also a vasodilator, improving blood flow to tissues, which is explored further in Section 5.4.

Whilst the exact mechanism by which NIR increases iNOS is unknown, it is established that the resultant NO plays a key role in PBMT that could be utilized in COVID-19 treatment. Due to its ability to reduce platelet activation, and the role of platelet adhesion in thrombosis, NO has further potential to treat thrombosis in COVID-19 patients [95,96].

### 5.3. Inflammation

One study [72] found that IR1072 treatment increased mRNA expression for cytokines responsible for the acute phase of the immune response (IL-1β, TNF- α, IL-6 and MCP-1). After 3–5 days, these levels returned down to control levels: a normalization that is necessary to sustain the immune response’s homeostasis. This would be useful to combat COVID-19, where the immune response is often delayed, but excessive. PBMT has significant advantages over corticosteroids which have been researched for their anti-inflammatory use against COVID-19 [97], including a lack of side-effects and no known interactions with the underlying conditions common in COVID-19 patients. Numerous studies have demonstrated that PBMT reduces pro-inflammatory cytokines and increases anti-inflammatory cytokines in in vivo models [97,98,99]. 

PBMT also shows multiple effects on reactive oxygen and nitrogen species (RONS). It appears that PBMT may decrease ROS in cells already undergoing oxidative stress, e.g., in animal disease models, but increases ROS production in normal, healthy cells [100]. The transcription factor NF-κB also shows contradictory behaviour with light irradiation, with two papers from a single laboratory showing both the activation [101] and inactivation [102] of NF-κB by 810 nm therapy. The authors postulated that NF-κB signalling is enhanced in normal, healthy cells treated with PBM, but is reduced when PBMT is applied to inflammatory cells with sufficient antioxidants. NF-κB up-regulates genes encoding pro-inflammatory cytokines, intensifying the inflammation. It may be that PBMT initially acts in a pro-inflammatory manner but, after a short period of time (a few days), gives the usually more desired anti-inflammatory response. This may provide ideal protection against the cytokine storm of COVID-19. An initial burst of RONS by light treatment could be a means of preconditioning the cells to oxidative stress, such as in viral infection. As the scientific focus for COVID-19 therapies shifts towards attenuating the patient’s inflammatory response, the use of PBMT is supported to reduce hospitalisations and deaths, especially related to the cytokine storm.

### 5.4. Blood Flow and Thrombosis

PBMT improves blood flow and, therefore, oxygen availability and consumption [75]. One mechanism by which PBMT enhances blood circulation is through the increase in vascular endothelial growth factor (VEGF): a protein that stimulates the formation and growth of blood vessels to increase oxygen supply [72]. The transcription of VEGF is regulated by hypoxia-inducible factor (HIF) 1-α which, in turn, is stabilised by the dissociation of NO from CCO [103]. The PBM-induced increase in NO (Section 5.2) and photo-activation of CCO is, therefore, able to stimulate blood flow, alongside the direct vasodilator action of NO. An increase in blood flow and vasodilation affects inflammation by increasing oxygen to the organ under oxidative stress, and by facilitating the transport of immune cells to the irradiated site, which is beneficial in an antiviral therapy as it allows for faster rehabilitation. A 1991 study of 60 preoperative oncology patients found that PBMT increased the total immune response, including changes to leucocytes, lymphocytes, monocytes, immunoglobulins, and active T-lymphocytes [104]. The photoactivation of the immunoresponse, alongside the photomodification of antigens, suggests that PBMT may be useful in an antimicrobial capacity, with the increased blood flow guiding the activated immune cells to the inflamed area.

Due to the rising body of evidence suggesting that COVID-19 may predispose thrombotic disease, there has been a global effort to prevent venous thromboembolism (VTE) in COVID-19 patients (during hospitalisation and after discharge) and to discover the ideal management of patients with both COVID-19 and VTE diagnoses. Some of the therapies under investigation for COVID-19 may pose distinct drug–drug interactions with common antithrombotic medications, highlighting the need for non-drug treatments for patients with COVID-19-induced thrombosis. Importantly, in a recent extracorporeal blood flow porcine study of PBMT, the aggregation of platelets in the control group increased throughout the 24 h post-operative period, whereas platelet aggregation in the 700–1100 nm group remained stable or decreased in intensity [105]. This shows the potential of PBMT to decrease the risk of fatalities from thrombosis by reducing aggregration. Furthermore, a 2008 study with a model of embolized rabbits explored the safety of combining the thrombolytic tissue plasminogen activator (tPA; Alteplase) and transcranial near-infrared laser therapy (TLT, i.e., PBMT). PBMT administration did not significantly affect the increase in hemorrhage incidence caused by tPA, and the combination treatment did not exacerbate haemolysis. Therefore, TLT may be administered safely either alone or in combination with tPA, because TLT has no effect on hemorrhage incidence or volume [106].

### 5.5. Photo-Preconditioning by Heat Shock Proteins

Chronic PBMT upregulates heat shock protein (Hsp) expression, in particular, Hsp70, which is vital for cytoprotection. Using an Alzheimer’s disease mouse model, Grillo et al. (2013) showed that a number of Hsps were regulated by chronic IR1072 treatment for 5 months [107]. Hsp60, Hsp70, Hsp105, and phosphorylated Hsp27 were all significantly (*p* < 0.05) increased in the PBMT compared to age-matched controls. De Filippis et al. (2019) also showed changes in Hsp70 with IR radiation, in this case, a Q-switch 1064 nm treatment to human keratinocytes [108]. Mechanisms of Hsp action may explain some of the previously discussed actions of PBMT, including the increased expression of proinflammatory cytokines [108]. Hsp70, alongside Hsp90 and Hsp27, upregulates proinflammatory cytokines IL-6 and TNF-α, and it was shown with rat and mouse microglial cells that this is through Toll-like receptor (TLR-) 4 activation [109]. Asea et al. (2002) demonstrate that Hsp70 induces cytokine production via MyD88 and NF-κB modulation [110].

Importantly for COVID-19 treatment, the synthesis and release of Hsp70 may have direct and transient antiviral effects. These Hsp70-induced antiviral effects have been seen in treatments for numerous viral infections, including influenza A virus [111], rhinovirus [112], HIV [113], and Sindbis virus [114]. The mechanism by which Hsps disrupt viral synthesis has yet to be discovered, although it is suggested that Hsp70 interferes on several levels by blocking transcription and/or translation. Hsp70 may prevent translation by interacting with nascent viral polypeptides directly, or by competing with the viral translation mechanisms [111,113]. The release of Hsp70 by virus-infected cells stimulates macrophage and microglia innate immune responses. A positive feedback loop is generated between viral gene expression in host neurons and extracellular Hsp70 release [115,116]. Extracellular Hsp70 acts as a damage-associated molecular pattern (DAMP) molecule, binding TLR2 and TLR4 [116,117,118]. This interaction stimulates signalling pathways involving interferon regulatory transcription factor (IRF-) 3 and NF-κB that go on to increase the expression of type 1 interferons (IFN-β in the brain) and antigens presenting complexes (major histocompatibility complexes; MHC) [116,119]. The expression of the antiviral cytokine IFN-β by macrophages in the brain is vital for neuronal immunity to some, but not all, viruses [120].

The importance of Hsp70 in preventing thrombus formation has also been recently shown. WT mice showed delayed thrombus formation, but unaltered tail bleeding time, when given the Hsp70 inducers TRC051384 and tubastatin A [121]. These inducers act through two different pathways, highlighting the specific role of Hsp70′s in preventing clots. Even when aspirin was given at the same time, Hsp70 inducers did not raise the risk of bleeding.

Hsps are increased with high temperatures, a common symptom in COVID-19 patients. This rise in chaperones may be beneficial for cytoprotection and as an anti-viral agent against SARS-CoV-2. However, the medications COVID-19 patients may take to decrease their temperature will reduce levels of Hsps. Therefore, PBMT in conjunction with these medications may be ideal, maintaining Hsp70 for its antiviral and protective effects whilst allowing bodily temperatures to return to normal physiological levels.

## 6. Conclusions

Despite a level of natural antiviral immunity, the high infection rate of COVID-19 suggests that this is not sufficient protection against SARS-CoV-2 infection for millions worldwide, especially for immunocompromised or fragile individuals. With 11.3 billion vaccine doses administered (https://covid19.who.int, accessed on 18 April 2022), and rapid and PCR tests allowing us to monitor cases and reduce infection spread, we are fortunately in a better position than in the earlier days of this pandemic. Nevertheless, we are still in the pandemic and, with high case numbers continuing, it remains imperative to find treatments options for those with COVID-19 complications, and for the inevitable coronavirus viral pandemics of the future. One such treatment is PBMT 1068 nm, which could be applied at both the early (acute infection) and late (long COVID) stages of COVID-19 to the nasal cavity or torso/lungs (or to the brain in the case of neuroprotection from neurological symptoms, or to the skin with respect to dermatological symptoms). It could be utilised for those with severe COVID-19, or as a preventative strategy in high-risk individuals who could benefit from PBMT when their disease is still in its early stages. Unlike immunosuppressants, PBMT does not cause a delay in the antiviral response [122], Moreover, the effects of PBMT are localised, without any adverse side effects or drug–drug interactions, and it acts to boost the body’s natural immune response.

Our findings indicate that PBMT 1068 nm is indeed a viable therapeutic option against COVID-19 viral infection and its complications, including the cytokine storm, ARDS, and thrombosis. As shown in pre-clinical studies, PBMT is able to treat acute lung injuries and ARDS though a reduction in pulmonary inflammation, an increase in oxygenation, and the regeneration of injured tissues. The molecular mechanisms of PBMT support this, with Figure 4 summarising the interconnected effects. We propose that NO and Hsp70 are major molecular players in the positive actions of PBMT 1068 nm through the prevention of coronavirus replication, the induction of vasodilation, and increases in blood flow and ATP, together with PBMT’s anti-inflammatory and anti-thrombotic actions. To objectively evaluate the efficacy and safety of the promising 1068 nm PBMT on COVID-19, randomised, double-blind, placebo-controlled clinical studies are recommended as soon as possible. 

## Figures and Tables

**Figure 1 ijms-23-05221-f001:**
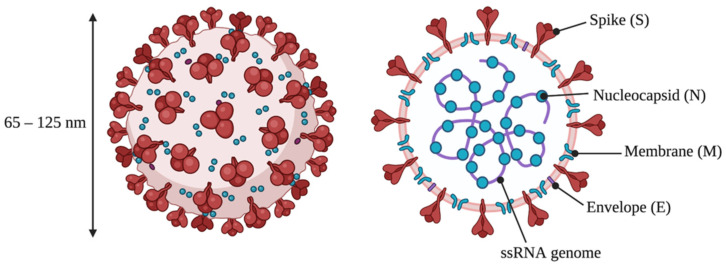
Structure of SARS-CoV-2, showing the four main structural proteins and the viral genome. Adapted from “Human Coronavirus Structure”, by BioRender.com (2022). Retrieved from https://app.biorender.com/biorender-templates (accessed on 13 January 2022).

**Figure 2 ijms-23-05221-f002:**
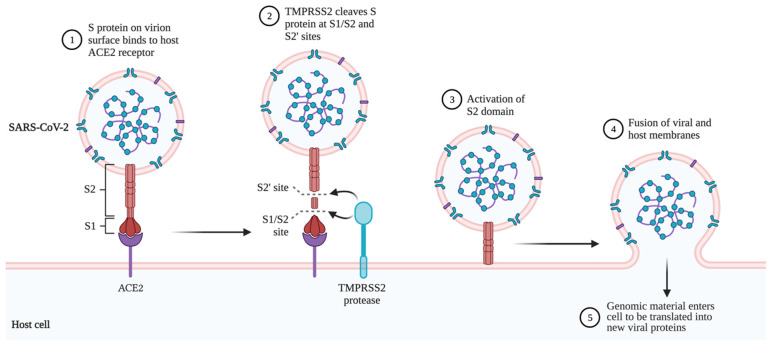
Mechanism of SARS-CoV-2 entry into a target cell. Created using BioRender.com (2022). Retrieved from https://app.biorender.com/biorender-templates (accessed on 13 January 2022).

**Figure 3 ijms-23-05221-f003:**
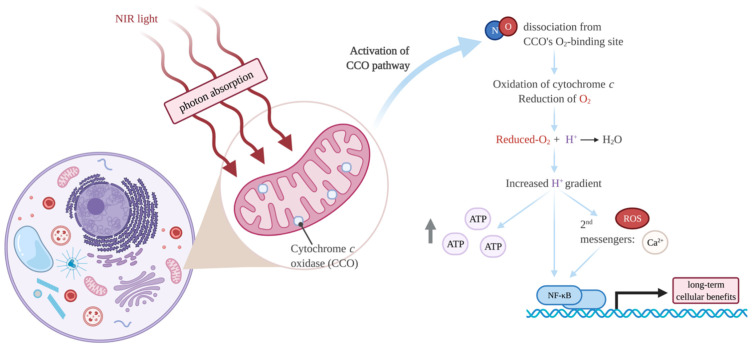
Cellular actions of photobiomodulation therapy (PBMT) through cytochrome *c* oxidase (CCO) activation by near-infrared (NIR) light. Created with BioRender.com (accessed on 13 January 2022).

**Figure 4 ijms-23-05221-f004:**
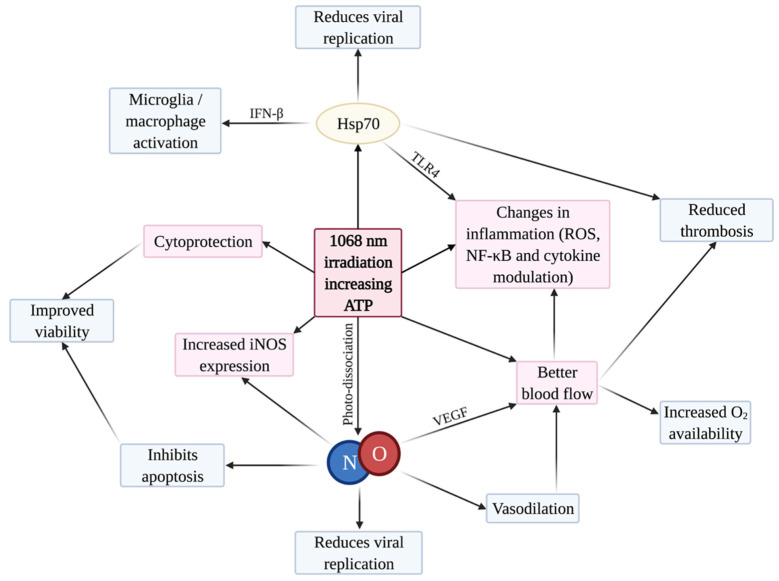
Summary network of PBMT effects, with a focus on antiviral mechanisms. Created with BioRender.com (accessed on 13 January 2022).

## Data Availability

Not applicable.

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
