# Peer review of "Rationale for 1068 nm Photobiomodulation Therapy (PBMT) as a Novel, Non-Invasive Treatment for COVID-19 and Other Coronaviruses: Roles of NO and Hsp70"

_ijms, 2022, doi:10.3390/ijms23095221_

Round 1

Reviewer 1 Report

This review paper focus on the PBMT, which wavelength is 1068nm, applied to COVID-19 viral infection. Overall, this paper is well-written and suitable for the scope of this journal. I would only provide some minor suggestions as followed:

  1. The immune system of the antibodies and lymphocytes induced by PBMT are lacked in this article. If the authors can improve or add it in this article, it would be better.

For example, in line 323-325, “This also guides immune cells such as macrophages and granulocytes to the irradiated area, which would be beneficial in an antimicrobial therapy.” I think parts of the experimental results published by Skobelkin (as the attached file) in 1991 should be cited to enhance the effects of PBMT on immune system. The photoactivation of the autoimmune system and the tumoural antigen photomodification have been seen in this research, including the antibody variations. May be the PBMT at 1068nm has the similar effects on immune system. If the authors can discuss it in the article, this article would be more completed.

  1. Line 204, the full name of ETC should be added, line 333, the full name of VTE, venous thromboembolism, should exist at first time.

Author Response

 Reviewer 1 Comments

“This review paper focus on the PBMT, which wavelength is 1068nm, applied to COVID-19 viral infection. Overall, this paper is well-written and suitable for the scope of this journal. I would only provide some minor suggestions as followed:

  1. The immune system of the antibodies and lymphocytes induced by PBMT are lacked in this article. If the authors can improve or add it in this article, it would be better.

For example, in line 323-325, “This also guides immune cells such as macrophages and granulocytes to the irradiated area, which would be beneficial in an antimicrobial therapy.” I think parts of the experimental results published by Skobelkin (as the attached file) in 1991 should be cited to enhance the effects of PBMT on immune system. The photoactivation of the autoimmune system and the tumoural antigen photomodification have been seen in this research, including the antibody variations. May be the PBMT at 1068nm has the similar effects on immune system. If the authors can discuss it in the article, this article would be more completed.

  1. Line 204, the full name of ETC should be added, line 333, the full name of VTE, venous thromboembolism, should exist at first time.”

We thank the reviewer for their kind remarks and for these suggestions, and we have made the appropriate amendments to the submitted manuscript.

Reviewer 2 Report

The manuscript “Rationale for 1068 nm photobiomodulation therapy (PBMT) as a novel, non-invasive treatment for COVID-19: roles of NO and Hsp70” by Lydia C. Kitchen et.al., summarizes a detailed research work on how photobiomodulation at 1068 nm could potentially be used as a  non-invasive treatment to investigate cellular & molecular mechanisms and understand these changes in people with COVID-19.

  • The authors also claimed that the pro-inflammatory and pro-thrombotic nature of COVID-19, photobio- modulation therapy (PBMT) could be used to modulate the immune and thrombotic system and repair damaged host tissues.
  • The authors also claimed that the specific use of 900-1068 nm near infrared (NIR) light could help in the treatment of patients infected by SARS-CoV-2, including direct and indirect antiviral effects.

From a scientific standpoint, I do understand that use of radiation or photobiomodulation at 1068 nm can help in non-invasive destruction of cells. But this is known to all. There is no novelty in this connecting it with COVID-19. Even when a person has a flu/cold or a viral fever, there are changes in the body cellular level, hence saying that those cellular changes/ non-invasive therapy at 1068 nm could help provide therapy after flu does not make any sense. The reason why I tell this, is because our human body produces antibodies against those viral antigens to help in self-recovery of the body. Why we need to do photobiomodulation at 1068 nm? Second reason, there are millions of people who had COVID-19, I doubt if any of those would volunteer for photobiomodulation therapy? The research community and the general community are interested in having easy to use kits to detect COVID-19, or an over-the counter drug or a flu shot to help combat flu or COVID-19.

I read your manuscript and from a scientific perspective its good but connecting it with COVID-19 makes no sense to me at least. I would recommend the authors to create a new storyline to present their manuscript.

Author Response

Reviewer 2 Comments

This rating does not fit with the refree comments below. Please clarify.

The manuscript “Rationale for 1068 nm photobiomodulation therapy (PBMT) as a novel, non-invasive treatment for COVID-19: roles of NO and Hsp70” by Lydia C. Kitchen et al., summarizes a detailed research work on how photobiomodulation at 1068 nm could potentially be used as a non-invasive treatment to investigate cellular & molecular mechanisms and understand these changes in people with COVID-19.

The authors also claimed that the pro-inflammatory and pro-thrombotic nature of COVID-19, photobio- modulation therapy (PBMT) could be used to modulate the immune and thrombotic system and repair damaged host tissues.

The authors also claimed that the specific use of 900-1068 nm near infrared (NIR) light could help in the treatment of patients infected by SARS-CoV-2, including direct and indirect antiviral effects.

We agree with these summary statements.

From a scientific standpoint, I do understand that use of radiation or photobiomodulation at 1068 nm can help in non-invasive destruction of cells. But this is known to all. There is no novelty in this connecting it with COVID-19.

We are unsure what the reviewer is referring to by “can help in non-invasive destruction of cells” Please clarify.

We have explained in detail the mechanisms underlying the action of the PBMT-T with the wavelength 1068 nm and propose that this would be useful as antiviral mechanism for coronaviruses, as seen in the successful treatment of other viruses such as Herpes.

The review covers both the specific wavelength 1068 nm and the neurological effects of COVID-19 (encephalopathy/brain fog etc), which is a completely novel concept, with no such publications in the literature. We would argue that PBMT is highly novel with potential as a safe and effective treatment for the whole-body effects of COVID-19 and other coronaviruses.

Even when a person has a flu/cold or a viral fever, there are changes in the body cellular level, hence saying that those cellular changes/ non-invasive therapy at 1068 nm could help provide therapy after flu does not make any sense.

We argue that this non-invasive safe therapy would be useful at all stages of viral infection: in the acute early stages of infection (in particular viral replication to reduce viral load), during infection with protection from cellular consequences such as inflammation/ thrombosis/ neuronal loss NB: IR1068 has been shown by the authors to be effective in treating dementia in early to late Alzheimers disease, via many of the mechanisms linked to CV19 (cited in review).

The reason why I tell this, is because our human body produces antibodies against those viral antigens to help in self-recovery of the body.

We are unsure of the relevance of this comment. We agree, there is a level of natural immunity against viral antigens indeed in many fortunate individuals. As evidenced by the high infection rate for millions worldwide, this is clearly not sufficient for protection against COVID-19, and immunocompromised individuals would argue that this natural immunity is not sufficient for protection from COVID-19, given the high death rate remaining in this fragile group. We would argue that infection rates are still high even now, despite wide-spread testing and vaccinations. Our ICU colleagues (and the patients in the ICU themselves!) would also say that COVID-19 is not over, despite high vaccination rates, and we are still officially in a pandemic according to the WHO. This is not to mention the billions across the globe who have not had a single vaccination to date, and therefore still remain vulnerable.

We have now added a paragraph into the review which acknowledges natural antiviral immunity and the relevance of PBMT to this stage of the pandemic.

Why we need to do photobiomodulation at 1068 nm?

We certainly still need effective and safe treatments for COVID-19; at present there are no ideal anti-viral drugs (many have serious side effects or drug-drug interactions, which limits their use in the many patients with co-morbidities) or other therapies, so we have proposed and explained with mechanistic support how PBMT-1068 offers a novel, effective and safe alternative, and warrants consideration for COVID-19 and subsequent inevitable Coronavirus pandemics in the near future.

Second reason, there are millions of people who had COVID-19, I doubt if any of those would volunteer for photobiomodulation therapy? The research community and the general community are interested in having easy to use kits to detect COVID-19, or an over-the counter drug or a flu shot to help combat flu or COVID-19.

We disagree with the reviewer. If the public was aware that this is a safe, effective and easy to use non-invasively (even at home) potential treatment for acute and long-COVID (the latter of which is a very serious long-term health issue, with no current effective therapeutics). We would strongly argue that the public would consider this new approach (hence the reason for importance of this current review). We particularly expect this to apply to immunocompromised and fragile/elderly individuals, given the eagerness we have personally seen from the public to become PBMT trial participants for various ageing afflictions. In addition, seeing multiple COVID-19 infections in an individual is still very common, despite 3-4 doses of vaccine, and viral mutations may begin to compromise efficacy in the future.

COVID-19 tests are a valuable resource: one which we are privileged to have in countries such as the UK to help reduce infection rates. They do not, however, do anything at all to benefit people who have contracted COVID-19, including those who go on to get serious complications as discussed in the manuscript. For high-risk, fragile individuals, or those with severe complications from COVID-19, the thing lacking (even in 2022 with rapid tests, PCR tests and vaccinations) is a treatment to reduce symptoms and boost the immune response, without adverse reactions to drugs.

I read your manuscript and from a scientific perspective its good but connecting it with COVID-19 makes no sense to me at least.

We thank the reviewer for the positive comments on the scientific content, and there appears to be no issues which we need to address regarding this.

For reasons both in this letter and in the review itself, we disagree with the statement that that the link between PBMT 1068 and COVID-19 “makes no sense”, especially if there are no issues with the scientific perspective. We ask the reviewer to reconsider their comments.

I would recommend the authors to create a new storyline to present their manuscript.

We have modified our manuscript to propose this novel strategy is worth considering for the still ongoing COVID-19 pandemic, and have extended its therapeutic scope for future inevitable coronavirus pandemics.

Round 2

Reviewer 2 Report

None.